

**Characterization of the Atlantic Water and Levantine Intermediate**
**Water in the Mediterranean Sea using Argo Float Data**
Fedele Giusy[1,*], Mauri Elena [1], Notarstefano Giulio [1] and Poulain Pierre Marie[1]
(1) National Institute of Oceanography and Applied Geophysics, OGS, 34010 Sgonico (TS), Italy
* Corresponding author (gfedele@inogs.it)

8       The Atlantic Water (AW) and Levantine Intermediate Water (LIW) are important water

masses that play a crucial role in the internal variability of the Mediterranean thermohaline
circulation. In particular, their variability and interaction, along with other water masses that
characterize the Mediterranean basin, such as the Western Mediterranean Deep Water (WMDW),
contribute to modify the Mediterranean Outflow through the Gibraltar Strait and hence may
influence the stability of the global thermohaline circulation.

14       This work aims to characterize the AW and LIW in the Mediterranean Sea, taking advantage

of the large observational dataset provided by Argo floats from 2001 to 2019. Using different
diagnostics, the AW and LIW were identified, highlighting the inter-basin variability and the
strong zonal gradient that characterize the two water masses in this marginal sea. Their temporal
variability was also investigated focusing on trends and spectral features which constitute an
important starting point to understand the mechanisms that are behind their variability. A clear
salinification and warming trend have characterized the AW and LIW in the last two decades
($\sim$0.007 and 0.008 yr$^{-1}$; 0.018 and 0.007 °C yr$^{-1}$, respectively). The salinity and temperature trends
found at subbasin scale are in good agreement with previous results. The strongest trends are
found in the Adriatic basin in both the AW and LIW properties. A subbasin dependent spectral
variability emerges in the AW and LIW salinity timeseries with peaks between 2 and 10 years.
**Keywords:** Argo, Atlantic Water, Interannual variability, Inter-basin variability, Levantine
Intermediate Water, Mediterranean Sea, Trends


**Acknowledgments**

This research was funded by the Italian Ministry of University and Research as part of the Argo-
Italy program.

## 1 Introduction

The Atlantic Water (AW) and Levantine Intermediate Water (LIW) play a central role in the internal variability of the Mediterranean thermohaline circulation, contributing to the dense water formation in this enclosed basin (Tsimplis et al., 2006). The variability and interaction of these two water masses modulate the Mediterranean outflow through the Gibraltar Strait, which plays an important role on the North Atlantic oceanic variability, and in turn to the stability of the global thermohaline circulation (e.g., Rahmstorf, 2006; Hernandez-Molina et al., 2014). Therefore, from a climatic point of view, it is relevant to characterize their main properties and monitor their variability, which are the main purpose of this paper.

Flowing in the Mediterranean Sea through the Gibraltar strait, the AW is less dense than the surrounding water masses and therefore it populates most of the Mediterranean surface layer. Its path is mainly driven by the Coriolis effect and by the complex topography that characterizes this region (Millot and Taupier-Letage 2005).

The LIW is the most voluminous water mass produced in the Mediterranean Sea (e.g., Skliris 2014; Lascaratos et al., 1993), and the saltiest water formed with a relatively high temperature at intermediate depths. It is formed in the Levantine subbasin, after which it is named, where one of the main formation sites is the Rhodes Gyre (e.g., Tsimplis et al., 2006; Kubin et al., 2019). The LIW strongly influences the thermohaline circulation, flowing at intermediate depths and then passing over the sills, exiting the Gibraltar Strait and modifying the Atlantic circulation (Millot and Taupier-Letage, 2005).

Several studies have been devoted to the analysis of the AW and LIW main features and variability, taking advantage of different indicators to identify and track these two water masses in the Mediterranean Sea. Among them, the AW and the LIW are usually referred to the minimum





and maximum salinity in the surface and intermediate layers of the water column, respectively
(e.g., Millot and Taupier-Letage 2005; Bergamasco and Malanotte-Rizzoli, 2010; Mauri et al.,
2019; Kokkini et al., 2019). However, different approaches can also be found in the literature. In
particular Millot (2014) associated the LIW to the maximum of the potential temperature vertical
gradient found in an intermediate water layer, while Bosse et al. (2015) identified the LIW in the
northwestern Mediterranean Sea with the maximum salinity value found between two potential
density values ($\sigma_\theta = [29.03 - 29.10] \frac{kg}{m^3}$), encompassing both temperature and salinity maxima
characterizing the LIW layer. The main findings related to the hydrological properties of these
two water masses are summarized below.
The AW enters in the Mediterranean Sea through the Gibraltar Strait, occupying the upper 200
m of depth with potential density, temperature and salinity annual mean values: $\sigma_\theta \cong$
$[26.5 - 27] \frac{kg}{m^3}, T \cong [14 - 16]°C, S \cong [36.0 - 36.5]$ respectively. The AW flowing at the
surface, continuously interacts with the atmosphere and is subject to evaporation and mixing with
the underlying water masses. Flowing eastward, it becomes denser and the minimum salinity core
sinks. Therefore, it can be capped by the surface mixed layer and less influenced by air-sea
interactions. Its properties and variability are also modified by the local eddies and by the river
discharges in the coastal regions. These mechanisms shape the AW, leading to an increase of
salinity from about 36.25 in the Gibraltar Strait to values around 39.2 in the Levantine Sea (e.g.,
Bergamasco and Malanotte-Rizzoli, 2010; Hayes et al., 2019). These values highlight strong AW
temperature and salinity gradients between the Western Mediterranean (WMED) and the Eastern
Mediterranean (EMED).
The properties of the LIW core in the WMED are commonly referred to the following ranges
of potential density, temperature, salinity and depth, respectively: $\sigma_\theta = [29 - 29.10] \frac{kg}{m^3}, T =$
$[13 - 14.2]°C, S = [38.4 - 38.8], D = [200 - 600]m$ (e.g. Millot, 2013; Hayes et al., 2019);
while in the EMED these properties span over different values: $\sigma_\theta = [28.85 - 29.15] \frac{kg}{m^3}, T =$
$[14.6 - 16.4]°C, S = [38.85 - 39.15], D = [150 - 400]m$ (e.g. Lascaratos et al., 1993; Hayes
et al., 2019). Therefore, moving westward, $T$ and $S$ decrease and the LIW sinks.


These studies provide a general view of the AW and LIW properties in the Mediterranean Sea,
highlighting a strong inter-basin variability of these water masses along their paths, which in turn
influences their temporal changes.
An example is given by a recent paper by Kassis and Korres (2020), which provides a detailed
view of the EMED hydrographic properties for the period 2004–2017 taking advantage of Argo
data. Exploring the water column from the surface down to 1500 m in seven different regions of
the EMED, they revealed a high inter-annual variability of the stored heat and salt over this region.
In this study, following a similar approach, we investigate the AW and LIW properties,
isolating their main characteristics and variability from the surrounding water masses, taking
advantage of several diagnostics discussed in section 2.2. Our work aims to provide a more
detailed view of the AW and LIW thermohaline properties over the last two-decades in most of
the Mediterranean Sea, investigating these two water masses through a subbasin approach, which
aims to emphasize how the processes that take place in each subbasin shape the water masses
properties. Their temporal variability is also studied, discussing the relative trends and spectral
features, which constitute an important starting point to understand the mechanisms that are
behind their variabilty.
In the frame of climate change studies, it is important to estimate possible impacts of AW and
LIW changes on the Mediterranean Climate, since this region is one of the most vulnerable
climate change hotspots (Giorgi 2006). In fact, changes in temperature and salinity can strongly
affect the marine system over the Mediterranean and related human activities.
Previous studies highlighted a clear salinification of the Mediterranean Sea over the past few
decades (e.g., Painter and Tsimplis 2003; Vargas -Yáñez et al., 2010; Schroeder et al., 2017;
Skliris et al., 2018) and a clear deep water warming trend after 1980s, which in literature is often
related to the Nile River damming and to the global warming (Vargas-Yáñez et al., 2010). Positive
temperature and salinity trends, oscillating between [0.0016÷0.0091] °C/yr and [0.0008÷0.001]
$yr^{-1}$, respectively, are found in the deep layer (below ~700 meters) between 1950 to 2005 (e.g.,
Bethoux et al., 1990; Rohling and Bryden, 1992; Millot et al., 2006; Vargas-Yáñez et al., 2010;
Borghini et al., 2014).




This observed salinification and warming are also found at intermediate depths in several
studies (e.g., Zu et al., 2014; Schroeder et al., 2017; Skliris et al., 2018), with ranges that depend
on the region of investigation. A clear salinity positive trend between 150-600 m is found in the
Mediterranean Sea by Skliris et al. (2018), analyzing the MEDATLAS data from 1950 to 2002
($\sim0.007\pm0.004$ yr$^{-1}$).
In contrast, heterogeneous temperature trends are found in the upper layer in different regions
(Painter and Tsimplis, 2003). This sensitivity of the trends to the area of interest, can be due to
several reasons, such as the changes in the large-scale atmospheric forcing of the Mediterranean
region, the river runoff which differ from one region to another, and to the data coverage over a
specific area (e.g., Painter and Tsimplis, 2003; Vargas -Yáñez et al., 2009; 2010). In this respect,
Vargas -Yáñez et al. (2009) highlighted that the scarcity of data makes trend estimations very
sensitive to the data postprocessing, comparing results from different studies dealing with the
same time period. Therefore, in order to reduce the uncertainty of the trend estimations, longer
and less sparse timeseries are needed.
In this respect, this work aims to provide an updated view of the temporal evolution and trends
of the AW and LIW, taking advantage of the large observational dataset provided by the MedArgo
Program (Poulain et al., 2007). It covers the water column from the surface down to $\sim$2000 m
over the entire Mediterranean basin from 2001 to 2019. The Mediterranean Sea has been widely
studied through the deployment of hundreds of Argo profiling floats (Argo 2020) in the last two
decades as part of various national, European and international programs (Wong et al., 2021) and
with the participation of different institutions. For these reasons, this dataset constitutes an optimal
observational framework to investigate the AW and LIW properties.
The dataset and the methods used in this study are described in section 2, the results are
presented in section 3, where the inter-basin and inter-annual variabilities of the AW and LIW in
the Mediterranean Sea are shown. The main conclusions are drawn in section 4.

**2    Data and method**
**2.1    Data**



In this work the AW and LIW properties in the Mediterranean Sea are investigated taking
advantage of the Argo float dataset, which consists of more than thirty thousand *T-S* profiles for
the period 2001–2019. Since 2001, the number of observations is generally increasing, reaching
a peak of 4188 profiles in 2015, mainly thanks to the combined efforts of national and
international Argo initiatives. The deployments of most Argo floats in the Mediterranean were
coordinated by the MedArgo regional center (Poulain et al., 2007). In the Mediterranean, the
cycling period is usually reduced to 5 days, and the maximum profiling depth is mostly 700 or
2000 m (Poulain et al., 2007). The floats are equipped with Sea-Bird Conductivity-Temperature-
Depth (CTD) sensors (model SBE41CP; www.seabird.com/sbe-41-argo-ctd/product-
details?id=54627907875) with accuracies of ±0.002° C, ±0.002 and ±2 dbar for temperature,
salinity and pressure, respectively. The data measured by the profilers are transmitted to satellites
(e.g., via the Iridium or Argos telemetry systems), then to ground receiving stations, processed
and real-time quality-controlled by the Argo Data Assembly Centres (https://www.euro-
argo.eu/Activities/Data-Management/Euro-Argo-Data-Centres), sent to the Global Data
Assembly Center and made available for free to users. The delayed-mode quality control applied
on pressure, temperature and salinity follows the guidelines described in the Argo Quality Control
Manual for CTD (e.g., Wong et al., 2021; Cabanes et al., 2016), in conjunction with other
procedures developed at regional level (Notarstefano and Poulain, 2008; Notarstefano and
Poulain, 2013) to check the salinity data and any potential drift of the conductivity sensor.
The analyses are performed in eight Mediterranean sub-basins following the climatological
areas defined by the EU/MEDARMEDATLAS II project
(http://nettuno.ogs.trieste.it/medar/climatologies/medz.html), emphasizing the processes that take
place in each sub-basin and modify the water mass properties. Fig. 1 shows the geographical
distribution of the Argo profiles from 2001 to 2019 in the eight sub-basins considered (Algerian,
Catalan, Ligurian, Tyrrhenian, Adriatic, Ionian, Cretan and Levantine). The Alboran, Aegean and
the Sicily Channel sub-basins are not analyzed in this work due to the scarcity of observations in
these areas.
Most sub-basins are well spatially covered, except for the Adriatic Sea, where the majority of
observations are concentrated in the South Adriatic Pit (SAP) and therefore it is important to keep
in mind that the results found for this region, are representative of the southern Adriatic Sea. The
SAP is an important deep water convection site in the Mediterranean Sea (e.g., Kokkini et al.,
2019; Mauri et al., 2021, Azzaro et al., 2012; Bensi et al., 2013) and therefore it is also considered
as a crucial area from a climatic perspective. The temporal distribution of the float data is different
in the various sub-basins: the longest time series are available in the Ionian, Cretan and Levantine
regions, with data from 2001 to 2019, followed by the Algerian, Ligurian and Tyrrhenian sub-
basins where data are available after 2003, and then by the Adriatic Sea with data only after 2009.
In this context, it is important to mention that the low density in space and time of the Argo
profiles induces uncertainties in the results, especially during the first years of the analyzed
period.

### 2.2 Methods

As discussed in the introduction, many indicators/characteristics have been adopted in
literature to track the AW and LIW in the Mediterranean Sea. Most of them consider, as best
indicator, the minimum/maximum salinity at surface/intermediate layer for the AW/LIW and
motivated us to follow a similar approach (e.g., Millot and Taupier-Letage, 2005; Bergamasco
and Malanotte-Rizzoli, 2010; Hayes et al., 2019; Lamer et al., 2019).
A preliminary step in this analysis was the post-processing: we first applied a time sub-
sampling on each profiler to obtain a more homogeneous dataset (Notarstefano and Poulain,
2009). This is applied to each float as follows: if the cycling period is 1 day or less, the profiles
are sub-sampled every 5 days; if the period is 2 or 3 days, they are sub-sampled every 6 days; and
if the period is 5 or 10 days, no subsampling is applied. Afterward, each profile was linearly
interpolated from the surface (0 m) to the bottom every 10 m to obtain comparable profiles; and
finally, a running filter with a 20 m window, was applied to the data along the depth axis, in order
to smooth any residual spike.



After the post-processing, we proceeded with the search of the minima/maxima salinity peaks
for the AW/LIW between 0-100/100-700 m for each profile. We define a peak as a data point that
is smaller/larger than its two neighboring samples for the AW/LIW, imposing a minimum
difference value of 0.01. If no peaks are found due to high vertical mixing, the profile is excluded.
The peak with the maximum prominence in the salinity field identifies the AW/LIW in each
profile and is used to identify the respective depth and temperature of the water mass core.
Moreover, if the minimum/maximum salinity is located at the profile endpoints between 0-
100/100-700 m, it is associated to the AW/LIW only if its closest value satisfies the minimum
difference condition.
The number of profiles counted in each subbasin therefore not only depends on the data
sampling over that region, but also on external processes acting on the layer of interest such as
the mixing activity due to eddies and/or air-sea interaction: the stronger is the mixing, the lower
is the number of profiles considered.
Once the AW and LIW core are identified in each profile from 2001 to 2019, the AW and LIW
timeseries were computed in each subbasin to analyze the low frequency variability (LFV) and
trends at interannual to decadal timescale over the available timeseries. In this respect, the high
frequency variability was filtered out, first by subtracting the mean seasonal cycle to the raw
timeseries, and then applying a median yearly average filter. This last step is needed since the
data are not homogeneous in time in every subbasin from 2001 to 2019, and therefore without it,
the seasonal variability can contaminate the estimation of the trends. The latter have been
computed using the linear least-squares method to fit a linear regression model to the data.
The AW and LIW inter-basin variabilities were analyzed taking advantage of the boxplot
approach applied to each parameter and region (Fig. 2). Inside each grey box, the black bold line
indicates the median, while the bottom and top edges of the box indicate the 25th and 75th
percentiles respectively, and the black dots show the mode of each distribution, which
corresponds to the maximum probability density function (PDF). The whiskers (black dashed line
out of the box) extend to the most extreme data points not considering the outliers at the 5%
significance level ($pvalue \leq 0.05$). In order to test the significance, the Student's $t$ distribution



219 was applied to each hydrological parameter in every sub-basin (Kreyszig and Erwin 1970). The

220 null hypothesis (that states that the population is normally distributed) is rejected with a 5% level

221 of statistical significance. This method is also applied to the timeseries trends.

222  In section 3.1 we often refer to the range and skewness of the distributions, that are the

223 difference between the upper and lower limits and the measure of the symmetry of the

224 distributions, respectively (including only the 5% significance values).

225  After the timeseries post-processing, in order to provide a time-evolving map of the spectral

226 features in each subbasin for the AW and LIW hydrological properties, the continuous wavelet

227 transform (CWT; Grossman et al., 1990) was applied to the 1-year moving averaged time series.

228 The time series were also linearly interpolated in order to fill in small gaps. In this respect, to

229 evaluate the significance of the wavelet transform we took advantage of the Matlab ASToolbox

230 provided by Conraria and Soares (2011) and available at

231 http://sites.google.com/site/aguiarconraria/joanasoares-wavelets, based on Monte Carlo

232 simulations (Berkowitz and Kilian, 2000). Wavelet power spectra were estimated.

233

## 3 Results and discussion

235  In this section, the AW and LIW properties are investigated in the eight Mediterranean regions

236 before mentioned, focusing both on their spatial and temporal variability. The analysis of the

237 trends and spectral features are also shown.

238

### 3.1 Inter-basin variability

240 (i)  AW

241  The hydrological properties of the AW core in eight sub-basins (Fig. 1) are shown in Figs. 2a,

242 b, c, providing a compact view of the AW inter-basin variability for each parameter using the

243 boxplot approach.

244  In the whole Mediterranean Sea, several profiles are ignored (Fig.2d) because the minimum

245 salinity peak is smoothed by the air-sea interaction and eddies especially during winter (here not

246 shown). About the 20-30% of the data in most of the subbasins is discarded, while this percentage





exceeds ~34% in the Levantine subbasins, where the AW core is strongly perturbed by internal
and external processes (Malanotte-Rizzoli et al., 2003). Despite the substantial quantity of
excluded profiles (Fig. 2d), we were able to identify in a robust way the AW core in the available
dataset, characterizing its main properties and variability.

Moving eastward, the AW salinity increases from ~36 to 39.5 (minimum and maximum

whiskers limits; Fig. 2a), since the surface salinity minimum is progressively smoothed by
horizontal mixing with surrounding saltier waters. In fact, as discussed by Font et al. (1998) the
AW minimum salinity is dependent on the different degrees of mixing due to its residence times.

In the Algerian sub-basin, the salinity range reaches the highest extension compared to the

other regions, probably due to the large baroclinic instability that produces high mesoscale
variability in the surface layer and horizontal mixing by strong eddies (Demirov and Pinardi

2007).

The AW salinity range is smaller in the Catalan, Ligurian and Tyrrhenian Seas, where similar

distributions are found both in terms of range and skewness (which is close to zero): the main
mode and the median have salinity of ~38. In the Adriatic Sea the distribution is probably skewed
toward higher values because a clear positive salinity trend is found (Fig. 3; Lipizer et al., 2014).
In the Adriatic, Ionian and Cretan Seas, the range is higher than the surrounding sub-basins: in
the Adriatic and Ionian Sea this could be associated to the Bimodal Oscillation System (BiOS),
and then to the reversal of the North Ionian Gyre (Rubino et al., 2020), while in the Cretan Sea
we speculate that it is caused by the sinking of the AW during winter. This is in agreement with
Schroeder (2019), where it is shown that in the Cretan Sea, the strong wind-induced evaporation
and heat loss during winter lead the AW transformation into salty and warm Cretan Intermediate
Water. The depths reached in the Cretan basin (Fig. 2c) seem to confirm this hypothesis.

The AW temperature is highly variable, ranging between ~5 and ~30 °C, with a wider range

in the Catalan and Adriatic regions (Fig. 2b), possibly due to the higher seasonal sea surface
temperature variability over these sub-basins (Shaltout and Omstedt 2014). The lowest
temperatures detected can be related to the freshwater fluxes in these regions. In this respect, an
episode that can be relevant for the AW distribution in the Adriatic Sea is the large river runoff



observed in 2014 by Kokkini et al. (2019), which caused a saline stratification for more than a
year. This episode is also captured by our analyses (Fig. 3). As observed for the AW salinity
mode, even the temperature mode shift toward higher values moving eastward in agreement with
the literature (Bergamasco and Malanotte-Rizzoli 2010). In the Algerian basin the AW
temperature mode is higher than it is in the Catalan subbasin: this can be due to the influence of
freshwater fluxes in the Catalan region and led by the high eddy activity over the Algerian region
(Escudier et al., 2016) led by the strong baroclinic instability already discussed for the salinity
field (Demirov and Pinardi 2007).

The depths of the AW core oscillate between 0 and 90 m with the main mode sinking eastward

(Fig. 2c). The distributions are all skewed toward lower depths, with the maximum PDF near
surface and a median shifting from 0 to 45 m moving eastward, indicating a clear sinking of the
AW along its pathway.

(ii)    LIW

In this section, the main hydrological properties of the LIW are analyzed in each sub-basin.

Comparing the percentages of the number of detected LIW cores related to the total profiles (Fig.
2h), it is evident that less convection occurs at these depths in most of the Mediterranean Sea, and
therefore less profiles are excluded compared to the AW. However more profiles are rejected in
the Adriatic and Levantine Seas, because strong mixing processes tend to smooth the LIW core
in the intermediate layer. In fact, as already mentioned, the SAP is an important deep water
convection site (Azzaro et al., 2012; Bensi et al., 2013; Kokkini et al., 2019), while the Levantine
basin is the region where the LIW formation occurs.

In the other sub-basins, more than the ~80% of profiles is considered, suggesting that

identifying the LIW is much easier than the AW, which is highly perturbed by external forcings.

Flowing away from the region of formation, the LIW interacts with the surrounding water

masses and becomes less salty; the salinity gradually drops from ~39.2 to ~38.4 from the
Levantine to the Ligurian subbasin and then the trend becomes almost flat in the Catalan and
Algerian regions (Fig.2e). The distributions are highly symmetric around the median and the



variability decreases flowing westward maybe because the LIW becomes deeper, sinking from
~100 to ~650 m (Fig. 2g). The highest salinity is reached in the Cretan basin, where the formation
of salty and warm Cretan Intermediate Water, caused by strong wind-induced evaporation and
heat loss during winter, influences the LIW properties and detection (Schroeder, 2019).
The LIW temperature decreases westward from ~18 to ~12.8 °C. The range is higher in the
EMED as also found for salinity, suggesting that over this region, the intrusion of warmer and
saltier surface waters due to convective processes characterizes the LIW formation (Fig. 2f;
Schroeder, 2019).
The sinking of the LIW flowing westward is shown in Fig. 2g, dropping from about 100 to
650 m (maximum whiskers values). The distributions tend to be symmetric in most of the
Mediterranean Sea, except for the Adriatic Sea, where a strong LIW bimodality in the depth
domain is found (in agreement with Kokkini et al., 2019), with two peaks located at ~190 and
~500 m respectively (here not shown); this behavior explains the big range that characterizes this
region. The investigation of the Adriatic bimodality is beyond the scope of this paper.

**3.2 Interannual variability**
In this section, the temporal variability of the AW and LIW in each sub-basin is studied
analyzing the 1-year moving average timeseries and the relative trends.
The results of this analysis are affected by the irregular spatial and temporal sampling of the
Argo floats. Time gaps in the data are found in the Catalan, Tyrrhenian and Cretan Seas (Fig. 3).
The missing data are both due to the lack of Argo float samplings and to the exclusion of the
profiles where no clear salinity peaks are found (especially during winter). Data in the Adriatic
Sea are available only after 2009, while the Ionian, Cretan and Levantine sub-basins have much
longer timeseries, with data covering the period from 2001 to 2019.

**3.2.1 Trends**
(i)  AW trends



The AW salinity temporal evolution is shown in Fig. 3, where significant trends (at 5% level of
significance) are found in each region (Table 1). Positive trends are clearly found in the EMED,
highlighting a clear salinification of the AW in the last two decades over most of the
Mediterranean Sea (~0.007±0.021 yr$^{-1}$; Table 1). Comparable positive salinity trends between 0-
150 m (~0.009±0.009 yr$^{-1}$) are also found in Skliris et al. (2018) where multi-decadal salinity
changes in the Mediterranean Sea are investigated taking advantage of the MEDATLAS database
(MEDAR Group 2002) consisting of temperature and salinity profiles in the Mediterranean from
1945 to 2002 (https://www.bodc.ac.uk/resources/inventories/edmed/report/4651/). A clear
meridional separation between the WMED and the EMED is found in the AW trends during the
observed period. In the EMED the AW becomes saltier, reaching significant trends in the
Tyrrhenian, Adriatic, Ionian and Cretan subbasins, whilst in the WMED, a strong negative trend
emerges in the Algerian and Catalan subbasins (Table 1). This freshening of the AW inflow could
be related to the observed rapid freshening of the North Atlantic Ocean (Dickson et al. 2002),
which causes are related to different phenomenon, included the accelerating Greenland melting
triggered by the global warming (Dukhovskoy et al., 2019). These findings seem in contradiction
with the results provided by Millot (2007), showing a salinification of the Mediterranean inflow,
obtained analyzing autonomous CTDs on the Moroccan shelf in the strait of Gibraltar in the period
2003-2007. Nevertheless, comparing Fig. 3 in Millot (2007) and Fig. 3 in this work, a similar
positive trend is captured in the Algerian sub-basin, in the same period; while extending the
analysis to a longer timeseries, a clear negative trend leads the AW variability at interannual to
decadal timescale. Opposite trends are found in the EMED, where the very strong increase in net
evaporation of ~8 to 12% over 1950-2010 (Skiliris et al., 2018) and the damning of the Nile River
(as projected by Nof, 1979) may have caused the AW salinification. The trends are steeper in the
Adriatic, Ionian and Cretan sub-basins, where the salinity increases with an order of magnitude
higher ($O[10^{-2}]$) and the largest increase is found in the Adriatic Sea (0.051 $yr^{-1}$). Here the
impact of the negative E-P anomalies and large river runoff observed by Kokkini et al. (2019)
around 2014 is well captured by the salinity time series. The results in the EMED are in good
agreement with Fig. 9 of Kassis and Korres (2020), where the yearly average salinity per depth
zone and per region between 2004-2017 are shown. Similarities in the observed trend in the Ionian
Sea ($0.012\ yr^{-1}$) are also found with Zu et al., 2014 ($0.011\ yr^{-1}$), where the Argo floats data
between 2004 and 2014 are analyzed.
In contrast with the above mentioned meridional salinity transition from negative to positive
salinity trends moving eastward, the temperatures show a more homogeneous pattern of
variability, highlighting a significant increase of the AW temperature averaged over the eight
analyzed sub-basins ($0.018\pm0.026$ °C/yr; Table 1), in agreement with a recent report from the
Copernicus Marine Service, showing a warming of the Sea Surface Temperature ~0.04 °C/yr for
the whole Mediterranean basin (Schuckmann et al., 2019). Interbasin changes between the
subbasins are instead linked to changes in the large-scale meteorological forcing of the
Mediterranean region (Painter and Tsimplis, 2003). As found for the salinity field, the sharper
increase is related to the Adriatic Sea (~0.093 °C/year), highlighting the presence of mechanisms
that enhance the trends over this region.
The AW depths time series (Fig. 5) show a heterogeneous trend in the Mediterranean Sea,
with significant negative values (the depth decreases) in the Algerian, Ionian and Levantine
subbasins, and positive in the Tyrrhenian and Cretan regions (Table 1), which reflects into a
tendency of the AW to become shallower, increasing the stratification at basin scale
($0.022\pm0.216$).

(ii)     LIW
The LIW temporal variability is hereafter analyzed. Fig. 6 shows the salinity changes from
2001 to 2019 in the eight subbasins considered. A positive trend is found in the whole
Mediterranean Sea at 5% level of significance, highlighting a salinification also at intermediate
depths of this enclosed basin over two decades (~$0.008\pm0.007$ yr$^{-1}$; Table 1). A similar positive
trend between 150-600 m is found by Skliris et al. (2018), in the MEDATLAS data from 1950 to
2002 (~$0.007\pm0.004$ yr$^{-1}$). The LIW properties vary less than the AW as expected (about an order
of magnitude less), since it lies at deeper depths where air-sea interactions play a minor role. The





strongest salinity increase is found in the Adriatic Sea (0.025 yr⁻¹), exceeding the trends in other
regions by one order of magnitude.

The LIW salinity positive trends over the Mediterranean Sea are also found by Zu et al. (2014),

which confirms the salinification of the basin at intermediate depths, as also observed at surface
in most of the analyzed regions. This suggests that the enhancement of the net evaporation over
the Mediterranean in the last decades, that was observed by Skiliris et al. (2018), may lead the
formation of saltier LIW in the EMED, and as consequence a mean positive salinity trend over
the whole basin.

Positive temperature trends (5% level of significance) are found in the whole Mediterranean

Sea except in the Cretan and Levantine sub-basins where the LIW becomes colder (5% level of
significance; Fig. 7). This therefore highlights a zonal separation at intermediate depths in the
temperature trends between the Ionian Sea and the more eastward regions. By visual inspection,
a decadal signal overlaps the warming trend in the Cretan and Levantine sub-basins and matches
the low frequency signal captured by the correspondent salinity timeseries. Peaks of salinity and
temperature are observed in 2010 in the Levantine basin and then reach the Cretan Sea in ~2011.
The same variability is discussed in Ozer et al. (2017) and explained in connection with the Ionian
Bimodal Oscillating System (BiOS). These maxima are in fact attributed to periods of
anticyclonic circulation in the north Ionian (2006-2009) and limited AW advection to the south-
eastern Levantine basin, referring to the study by Artale et al. (2006). The LIW temperature mean
trend and standard deviation averaged over the eight subbasins are ~0.007±0.007 °C/yr (Table
1), which can be interpreted as a weaker response of the intermediate layers to the warming trend
observed at surface.

The sub-basins with the steepest increase are located in the central longitudinal band of the

Mediterranean Sea, therefore far from the LIW main sources. The Adriatic Sea has the larger
slope (0.025 °C/yr), followed by the Tyrrhenian (0.009 °C/yr) and the Ligurian, Ionian and Cretan
sub-basin with same mean trend (0.006 °C/yr). The range of temperature and salinity and the
respective variability in the Tyrrhenian and Ionian sub-basins are in good agreement with Poulain
et al. (2009), where *T* and *S* timeseries from 2001 to 2009 are computed from Argo floats data



near 600 m. The ranges and trends for *T* and *S* found in the Ligurian Sea are also confirmed by
Margirier et al. (2020), where vertical profiles collected by gliders, Argo floats, CTDs and XBTs
in the northwestern Mediterranean Sea over the 2007–2017 period are analyzed.

The LIW depth time series are shown in Fig. 8: significant negative trends (the depth

decreases) are found in the Tyrrhenian and Ionian Seas, while in the Adriatic and Levantine sub-
basins the LIW sinks ($pvalue \leq 0.05$). Non-significant trends are found in the other regions.
Abrupt shifts are found in the Adriatic sub-basin from ~200 m to ~500-600 m at different time
steps (trend ~26.397 m/yr), highlighting a bimodal behavior of the LIW depth and an intense
dense water production activity as also shown by Kokkini et al. (2019). Previous studies attribute
dramatic shifts in the Adriatic hydrological properties to the BiOS and the Eastern Mediterranean
Transient (e.g., Vilibić et al., 2012). This hypothesis can also be supported by correlations
between the BiOS (definition by Vilibić et al., 2020) and the AW/LIW salinity yearly averaged
timeseries in the Adriatic Sea, which maximum values are about -0.48/-0.46 at lag 0 /-4 yr (at
negative year lag, the BiOS leads; $pvalue \leq 0.05$). Further investigations are left to future
studies.

The results related to the EMED match those shown in Kassis and Korres (2020), where the

timeseries of salinity and temperature averaged between different depths-layers (below 100 m) in
similar subbasins are shown (see Fig. 8 in Kassis and Korres 2020). The LIW depth mean trend
and standard deviation averaged over the eight subbasins is 2.468±9.876 m/yr (Table 1).

**3.2.2 Spectral Analysis**

The AW and LIW salinity trends confirm a salinification of the Mediterranean Sea in the last

two decades. The causes behind these trends are still under debate and then deserve more
investigations. In this section, we take a first step in this direction, analyzing the spectral features
of the AW and LIW salinity yearly filtered time series (Figs.3-6).

**(a) AW**



The wavelet power spectra show that the AW salinity (Fig. 9), during the observed periods, is
driven by mechanism that acts at inter-annual timescale, with significant strong peaks (magnitude
higher than 0.5) oscillating between 2-6 years in each subbasin. A decadal variability ($pvalue \leq$
0.05) emerges in the whole Mediterranean Sea, except in the Adriatic and Ionian regions, with
stronger energy magnitudes in the Levantine and Tyrrhenian sub-basins. It is worth to mention
that these results are strongly time-length dependent: the longer is the time series, the higher is
the probability to capture a decadal signal if present; therefore, in this respect, the Adriatic time
series is penalized. Similar patterns are shared by the Ligurian and Tyrrhenian regions, with a
high persistent peak around 3 years between the years 2005-2015; while in the Ionian and Cretan
sub-basins the oscillations shift in ~2009 from periods of about 5 years to higher frequencies (~2.5
years).  In 2009, a strong peak at ~2-years is found in the Algerian, Cretan and Levantine sub-
basins, which overlaps a weaker decadal signal. As found in the salinity CWTs, the relative
patterns in the temperature domain highlight peaks between 2- and 5-years periods, while in the
depth domain strong peaks are centered around ~3 years in each region (not shown).

**(b) LIW**
Analyzing the LIW salinity CWTs (Fig. 10), a strong basin-dependent spectral variability
emerges. In this respect, in the Algerian, Ligurian, Ionian and Cretan sub-basins, the salinity is
mainly led by oscillations around 3-years, while in the Catalan, Tyrrhenian and Adriatic Seas the
period of the oscillations increases reaching ~5-8- years. Decadal oscillations dominate the
salinity variability in the Levantine region. In order to better investigate the decadal variability,
longer time series are needed.
A significant energy peak is found in the Algerian sub-basin before 2011 at period of ~2.5-3
years, which is also present in the Ligurian, Ionian and Cretan sub-basin in the next years.
Comparing the salinity CWTs for the AW and LIW in each sub-basin, other features emerge.
Strong differences appear in the Ligurian, Tyrrhenian, Ionian, Cretan and Levantine sub-basins:
in the Ligurian, Ionian and Cretan Seas, the low-frequency variability observed in the AW
disappears at intermediate depths, while in the Tyrrhenian and Levantine Seas the LIW does not



show high-frequency oscillations with period of ~2-3 years, suggesting that the AW is much more
non-stationary compared to the LIW. In fact, lying at deeper depths, it is less modulated by
external forcings (the seasonality, the air-sea interactions and the freshwater fluxes play a minor
role).

**4   Conclusions**
We presented an analysis of the main properties and variability of the AW and LIW in the
Mediterranean Sea, exploiting the Argo float data that provide an optimal observational dataset
to study their thermohaline properties. Indeed, this dataset covers the water column down to
~2000 m and provide data for almost two decades.
Taking advantage of different diagnostics discussed in section 2, the AW and LIW have been
detected in the Mediterranean Sea through a sub-basin approach, which allowed to define the
main hydrological features over this enclosed basin in different regions.
In addition to previous studies, this work provides a more detailed view of the AW and LIW
characteristics in the last two-decades over most of the Mediterranean Sea, except for the Alboran
sub-basin, the Sicily Channel and the Aegean sub-basin where Argo data are too scarce.
To achieve this goal, the first step of this study was the detection of the AW and LIW cores in
each available profile. In agreement with previous studies, we confirmed the zonal gradients of
the AW and LIW properties over the Mediterranean Sea: the AW becomes saltier, warmer, denser
and deeper moving eastward, while the LIW becomes less salty, colder, denser and deeper moving
westward. These results not only match the present literature but also provide a more detailed
view of these water masses over eight sub-basins.
The timeseries derived from the AW and LIW parameters have also highlighted some
interesting features that are in good agreement with the previous literature. The most relevant
results are summarized below:
•   Positive salinity and temperature trends characterize the AW and LIW in the last two

decades over most of the Mediterranean Sea (average value over the whole region:

0.007 and 0.008 yr$^{-1}$; 0.018 and 0.007 °C/yr respectively). The warming and

salinification of the Mediterranean Sea is in good agreement with previous results
(e.g., Skliris et al., 2018; Margirier et al., 2020; Kassis and Korres, 2020).
• Negative AW salinity trends in the Algerian and Catalan sub-basins suggest a
freshening of the AW inflow, in agreement with the observed rapid freshening of the
North Atlantic Ocean (Dickson et al., 2002).
• Positive AW salinity trends are found east of the Catalan sub-basin, highlighting a
clear salinification of this water mass in the last two decades probably due to the
combined effect of the strong increase in net evaporation and the Nile dumping (e.g.,
Nof, 1979; Skiliris et al., 2018; section 3.2.1a).
• Positive trends in the LIW salinity timeseries are found in the whole Mediterranean
Sea at 5% level of significance, highlighting a salinification also at intermediate
depths (section 3.2.1b).
• Positive LIW temperature trends ($pvalue \leq 0.05$) are found everywhere except in the
Cretan and Levantine sub-basins where the LIW becomes colder ($pvalue \leq 0.05$).
This highlights a meridional separation at intermediate depths in the temperature
trends between the Ionian Sea and the more eastward regions.
• The AW and LIW depth trends are highly space-dependent, showing different
behaviors in the eight sub-basins.
• The steepest trends are found in the Adriatic Sea in both the AW and LIW and for
each variable (Table 1). The AW temperature increases 0.093 °C/yr, while the LIW
temperature shows a trend about 0.059 °C/yr. Abrupt shifts are found in the Adriatic
sub-basin from ~200 m to ~500-600 m at different time steps (trend ~24.671 m/yr),
highlighting a bimodal behavior of the LIW depth and an intense dense water
production activity as also shown by Kokkini et al. (2019).
These results therefore provide interesting new insights about the AW and LIW interbasin and
interannual variability, which can be further analyzed to investigate which mechanisms lead to
the observed temporal trends in each sub-basin. A preliminary attempt in this direction is provided
by the spectral analysis of the filtered salinity timeseries (without seasonal variability). Peaks



between 2.5-6 years are found in the whole Mediterranean Sea in the AW salinity, and hints of
decadal variability appear everywhere, except in the Ionian and Adriatic regions. A strong basin-
dependent spectral variability emerges in the LIW salinity timeseries. In this respect, in the
Algerian, Ligurian, Ionian and Cretan sub-basins, the salinity is mainly characterized by
oscillations with a period around 3-years, while in the Catalan, Tyrrhenian and Adriatic Seas the
period of the oscillations increases to ~5-8- years. Decadal oscillations dominate the salinity
variability in the Levantine region, that could be related to the BiOS as suggested by Ozer et al.
(2017). In order to investigate the decadal oscillations, longer timeseries are needed. Finally,
comparing the AW and LIW salinity CWTs we find that the AW is much more non-stationary
compared to the LIW since flowing in the surface layer, it can be modulated by more external
forcings. Further studies on the leading mechanisms over each subbasin are left for future
investigations.



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




**Table 1.** Trends by year for the AW and LIW salinity (S), temperature (T) and depth (D) timeseries in eight Mediterranean subbasins. In bold characters the trends significant at 5% level. The rightmost column shows the mean and standard deviation trend values computed over the eight subbasins.

| TREND (S) 1/yr | Algerian subbasin | Catalan subbasin | Ligurian subbasin | Tyrrhenian subbasin | Adriatic subbasin | Ionian subbasin | Cretan subbasin | Levantine subbasin | Mean∓Std |
|---|---|---|---|---|---|---|---|---|---|
| AW | **-0.021** | **-0.004** | **~0.000** | **0.007** | **0.051** | **0.012** | **0.011** | 0.003 | $0.007 \pm 0.021$ |
| LIW | **0.003** | **0.003** | **0.006** | **0.009** | **0.025** | **0.006** | **0.006** | **0.002** | $0.008\pm0.007$ |
| **TREND (T) °C/yr** | | | | | | | | | |
| AW | **0.067** | **0.037** | **0.005** | **0.039** | **0.093** | **0.028** | **-0.024** | **0.033** | $0.018\pm0.026$ |
| LIW | **0.017** | **0.013** | **0.024** | **0.037** | **0.059** | **0.027** | **-0.012** | **-0.024** | $0.007\pm0.007$ |
| **TREND (D) m/yr** | | | | | | | | | |
| AW | **-0.135** | **-0.001** | **0.090** | **0.256** | **0.055** | **-0.277** | **0.361** | **-0.170** | $0.022\pm0.216$ |
| LIW | **-0.176** | **0.317** | **0.248** | **-5.630** | **24.671** | **-2.415** | **-0.426** | **1.213** | $2.225\pm9.323$ |






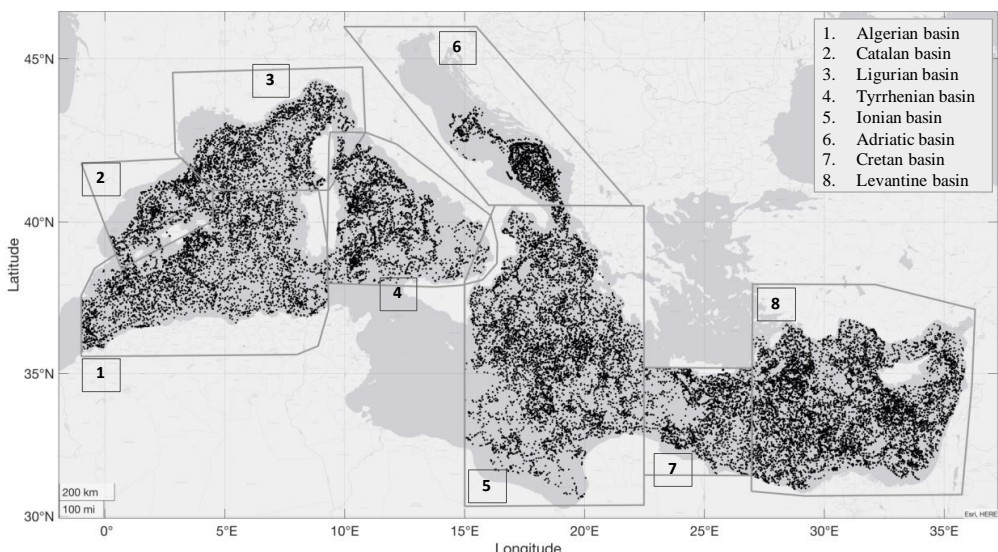


Fig. 1 Argo floats profiles scatter plot in the Mediterranean Sea between 2001 and 2019 in eight

regions based on the climatological areas defined by the EU/MEDARMEDATLAS II project.



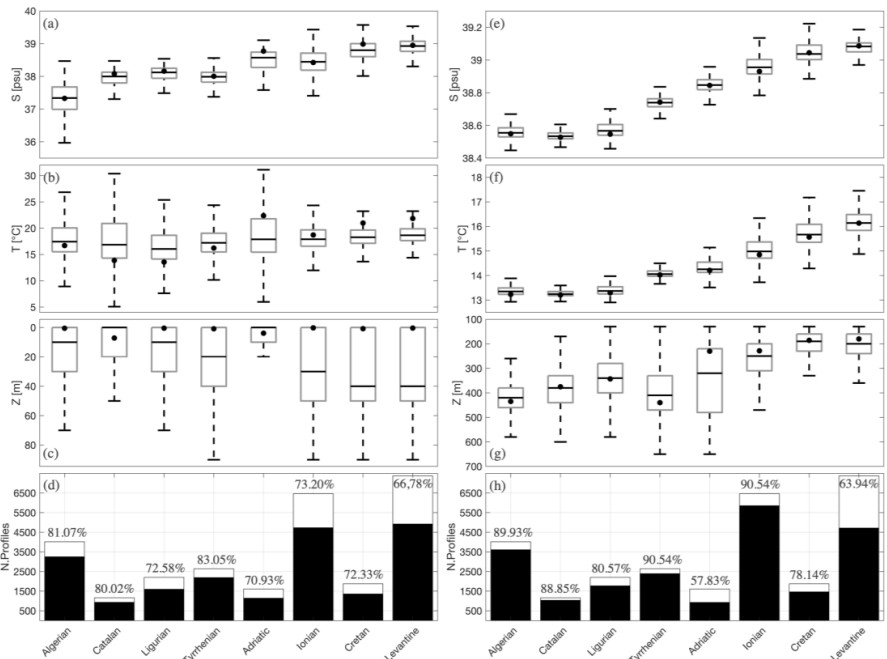

Fig. 2 Boxplot diagrams for the AW salinity (a), temperature (b) and depth (c) in eight
Mediterranean subbasins. The number of effective profiles (black bars) for each, compared with
the total profiles (white bars) are shown in panel (d). The numbers indicate the percentages of
effective profiles related to the total number available in each subbasin. The corresponding
diagrams for the LIW are shown in the panels (e,f,g,h).

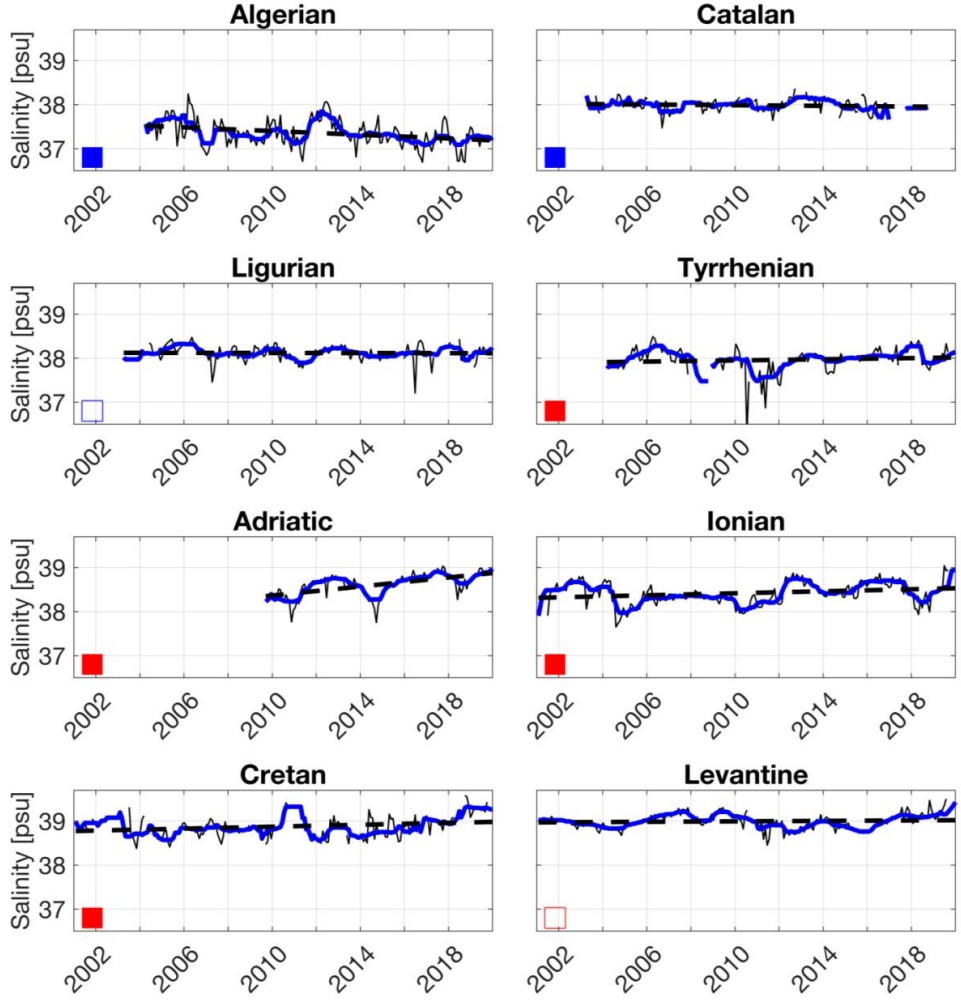

795

Fig. 3 AW salinity timeseries in eight subbasins: the thin black lines show the monthly timeseries

(seasonal cycle filtered out), the tick blue lines are the 1-year moving average timeseries and the

dashed black lines are the trends. The red/blue filled squares identify the positive/negative trends

with pvalue≤0.05, while the red/blue not-filled squares identify the positive/negative trends with

pvalue>0.05.



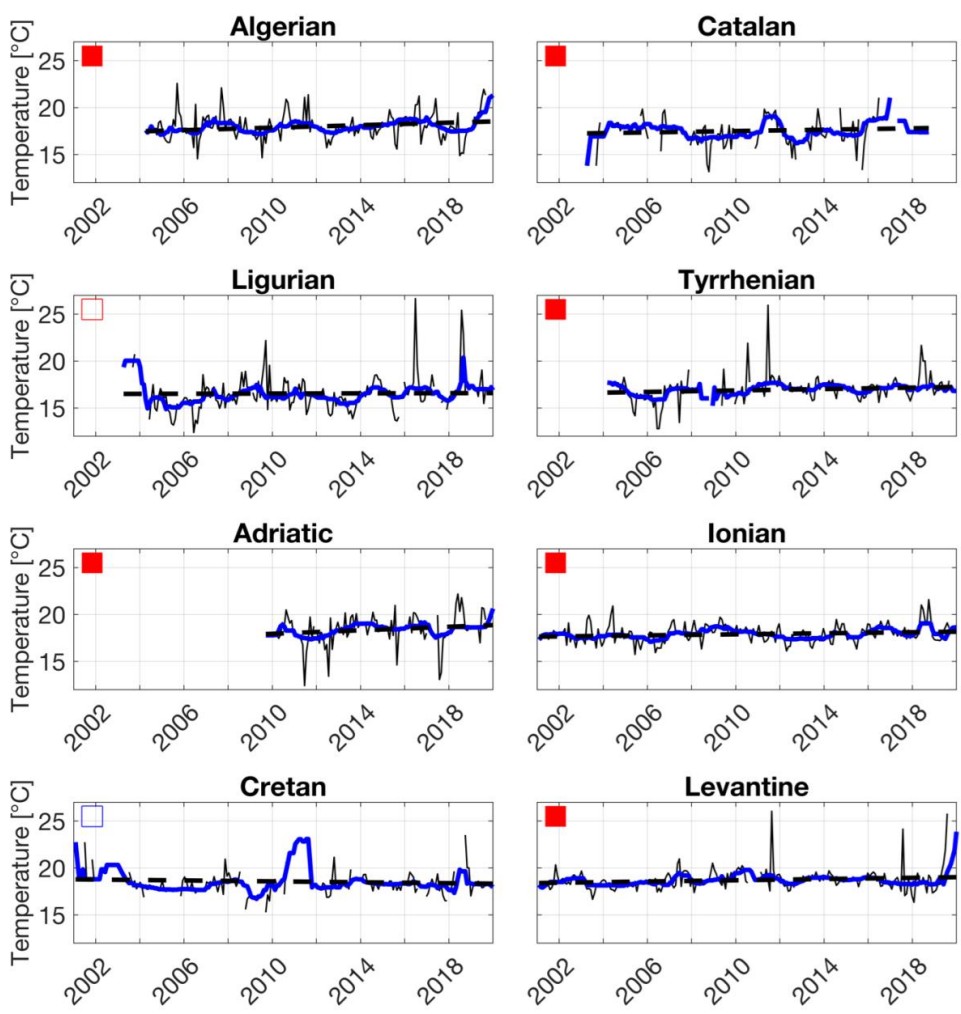


Fig. 4 Same as Fig. 3 but for the AW temperature.


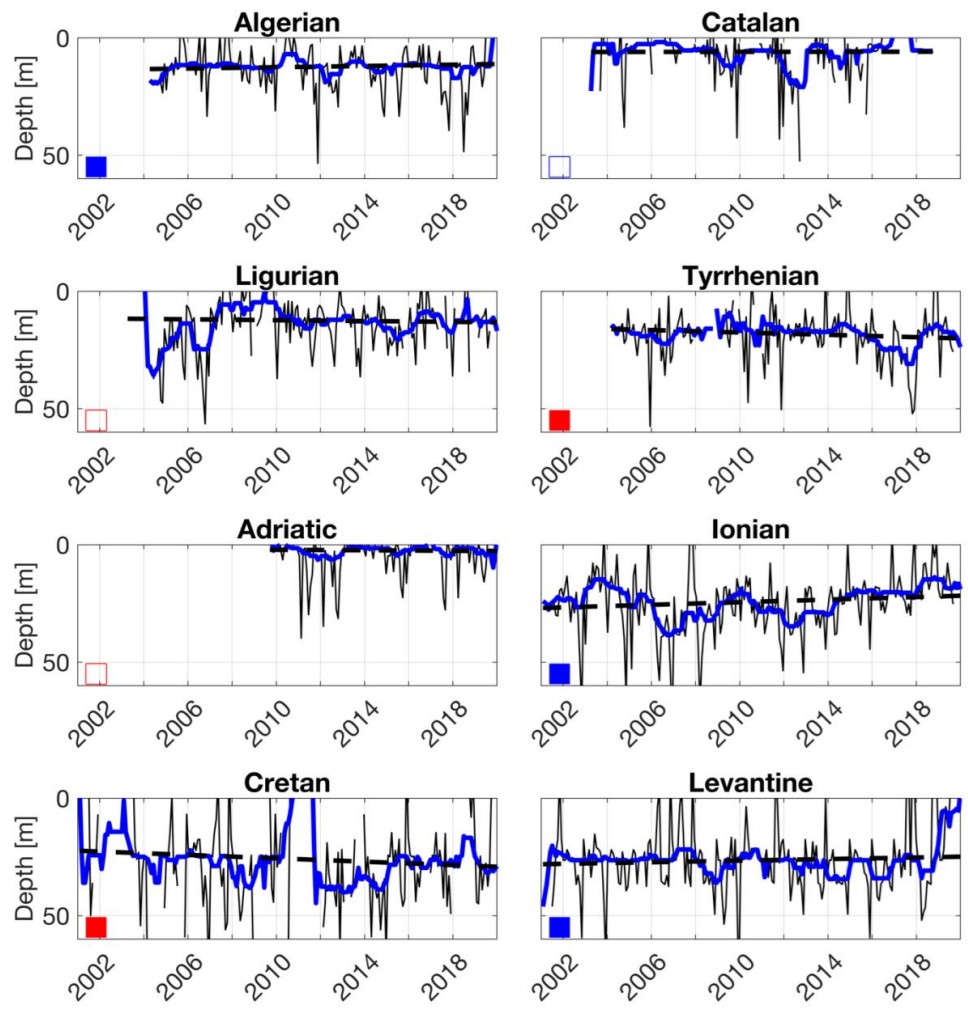

Fig. 5 Same as Fig. 3 but for the AW depth.



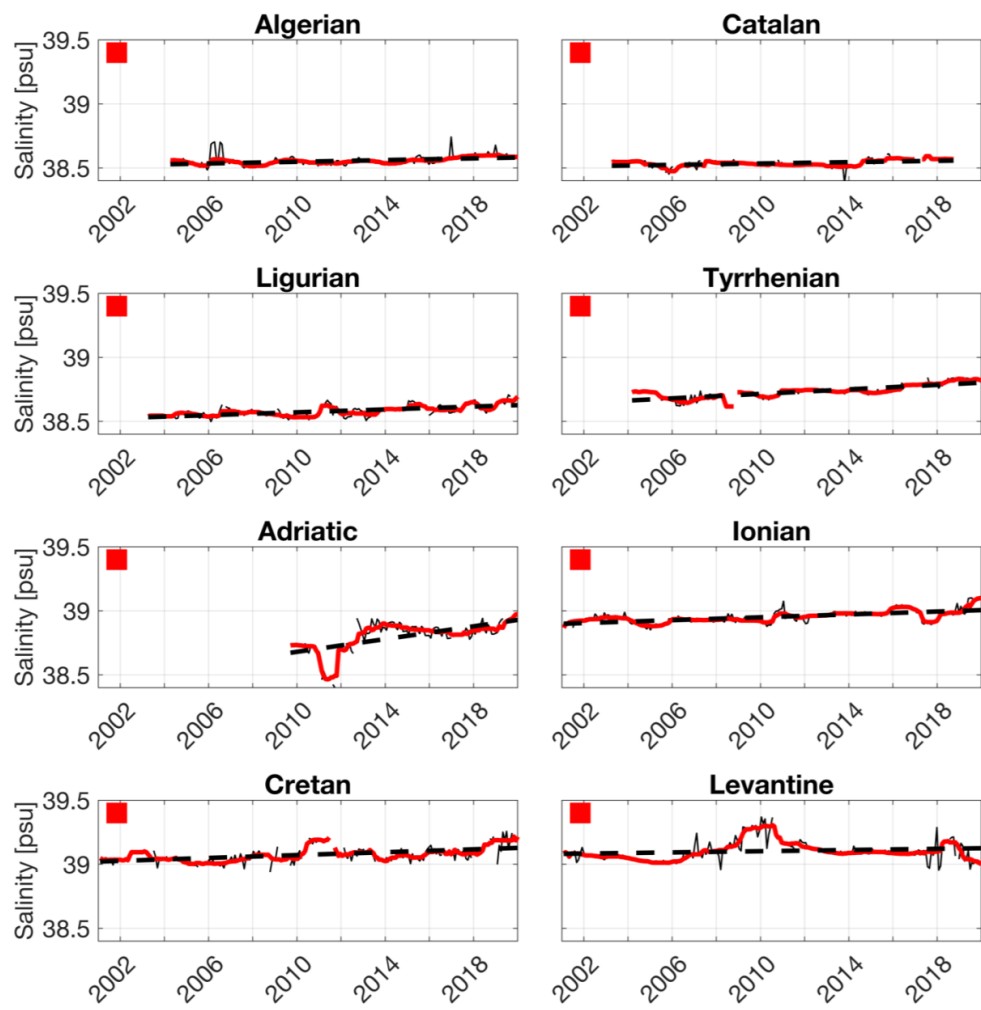


Fig. 6 Same as Fig. 3 but for the LIW salinity.




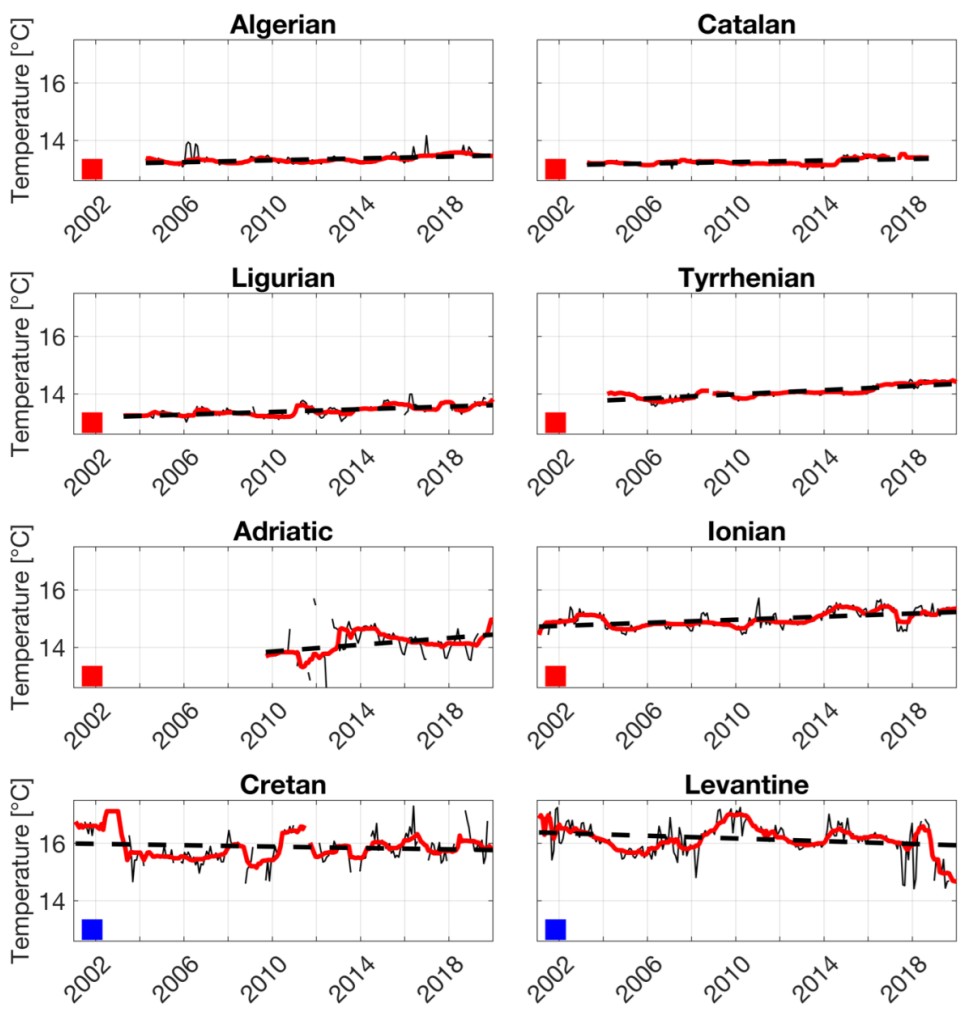


Fig. 7 Same as Fig. 3 but for the LIW temperature.





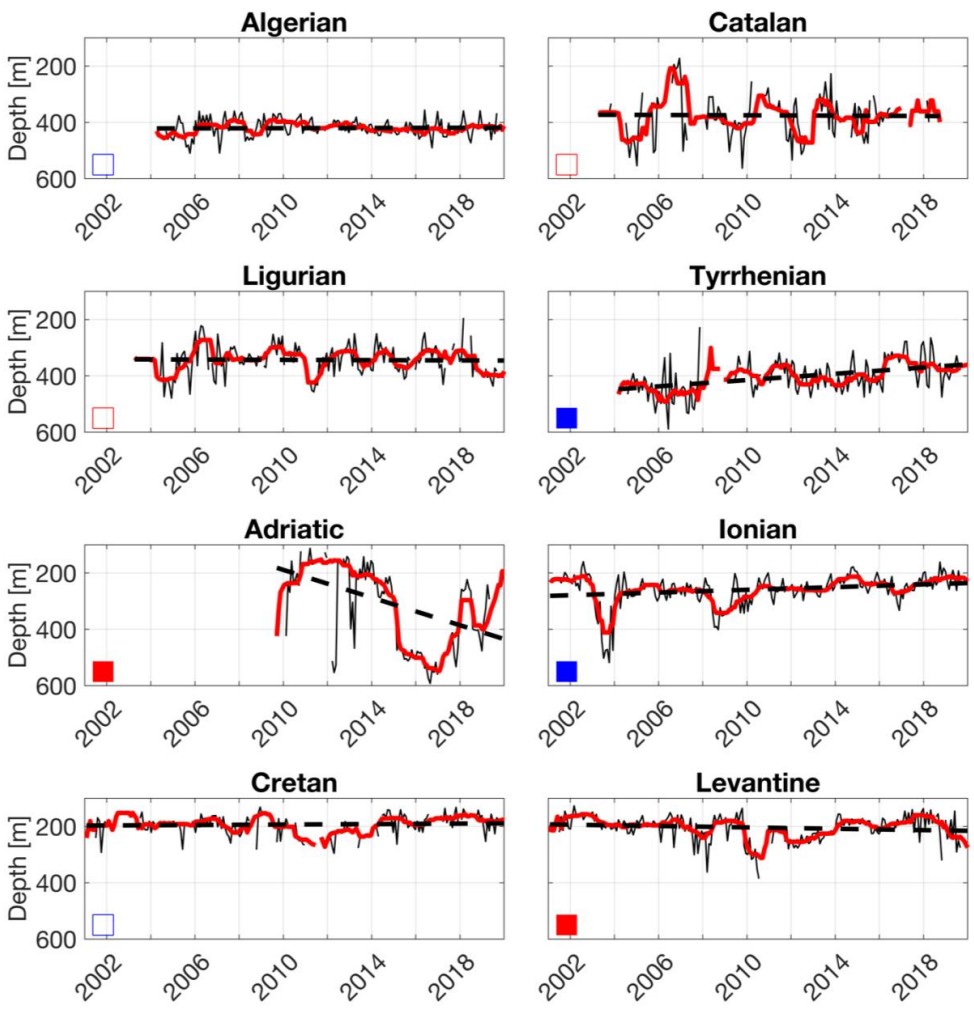


Fig. 8 Same as Fig. 3 but for the LIW depth.


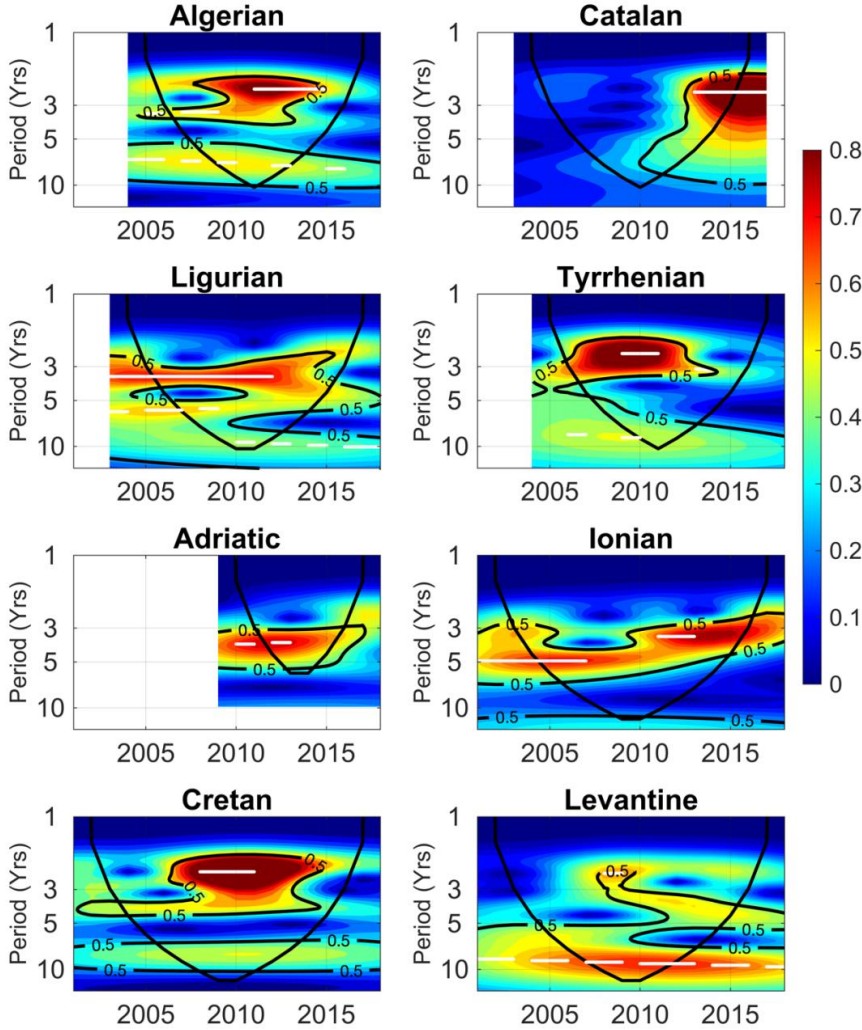


Fig. 9 Wavelet power spectra of the low-pass filtered AW salinity for the eight subbasins
considered. The black contour designates the 5% significance level based on Monte Carlo
simulations (Berkowitz and Kilian, 2000). The cone of influence, which indicates the region
affected by edge effects, is shown with a thick black line. The color code for power ranges from
blue (low power) to red (high power). The white lines show the main ridges of the wavelet power
spectrum.




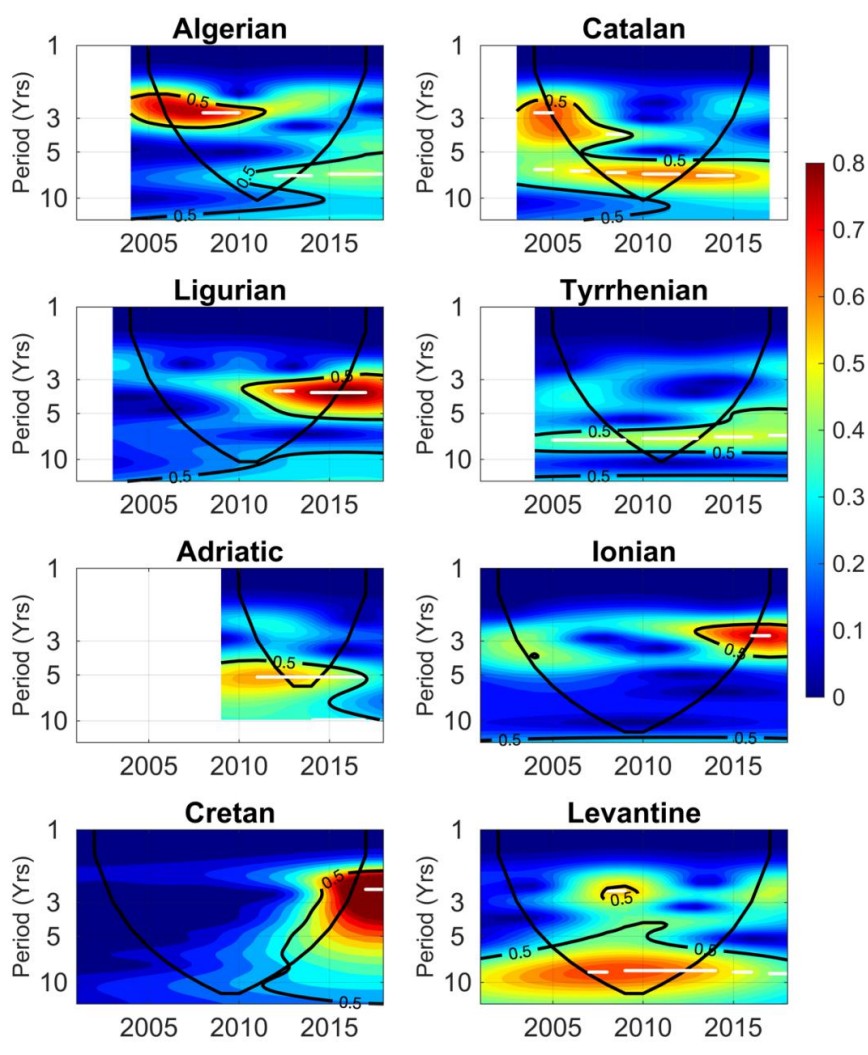


Fig. 10 Same as Fig. 9 but for the LIW salinity.