# Peer review of "Characterization of the Atlantic Water and Levantine Intermediate"

_Ocean Science, 2021_

## Author Response (AR1)

**AUTHOR'S RESPONSE**

We thank the two anonymous reviewers for the comments, which contributed to improve the quality of this paper. A detailed answer to the reviewer is provided below. In bold the reviewers' comments are shown. The updates in the manuscripts are in red font.

**REVIEWER#1**

**L49 I don't think Millot and TL is the appropriate reference here, you should replace it with one concerning the modification of Atlantic circulation due to the MOW**

The reference has been substituted with Rahmstorf, 1998; Bethoux et al., 1999, which are more appropriate than Millot and Taupier-Letage, 2005.

**L65 add references for these values**

The following references have been included Bergamasco and Malanotte-Rizzoli, 2010; Hayes et al., 2019.

**L98 write climate with low-case "c"**

This change has been applied.

**L111 should be "a clear positive salinity trend"**

This change has been applied.

**L138 write "30 000"**

We thank the reviewer for this suggestion, but we prefer "thirty thousand" more than 30 000.

L191-203 concerning the method of defining the AW and LIW, I have concerns that using the same method in the Alboran and in the Levant might give artifact results (same method and same depth ranges). These concerns are also about the same value of prominence (0.01) that you use for both water masses and in all regions. These concerns found justification later on, when I read that there are regions where you were forced to remove a very high amount of available profiles (up to 40%), which for me demonstrates that the method is not adequate

enough. The profiles that were removed in fact for sure contained AW/LIW in a pattern that should suggest you to modify your criteria (either depth range, or prominence value, or something else). You should also show a seasonal distribution, per subbasin, of the remaining profiles.

We thank the reviewer for this comment and take advantage of it to make more robust this study. The method previously shown aimed to capture the AW and LIW cores, reducing the "noise" provided by external factors. The method considered a unique threshold for each profile and basin, provided by the prominence value 0.01. We agree with the reviewer that using different prominence values for each basin would impact less on the number of profiles considered and as consequence on the results, but the definition of the best prominence for each region would need a more in-depth study, which is not the purpose of this paper. Therefore, we choose that arbitrary value, which excluded the profiles with a less clear minimum/maximum salinity value for the AW and LIW respectively, rejecting a smaller number of profiles than other prominence values. Nevertheless, we agree that this method is too arbitrary and less robust. Therefore, we decided to modify the method, following a new strategy, discussed below. The manuscript has been updated as follows. Please see lines L183 – L213.

"A preliminary step in this analysis was the post-processing: we first applied a time sub-sampling on each profiler to obtain a more homogeneous dataset (Notarstefano and Poulain, 2009). This is applied to each float as follows: if the cycling period is 1 day or less, the profiles are sub-sampled every 5 days; if the period is 2 or 3 days, they are sub-sampled every 6 days; and if the period is 5 or 10 days, no subsampling is applied. Afterward, each profile was linearly interpolated from the surface (0 m) to the bottom every 10 m to obtain comparable profiles; and finally, a running filter with a 20 m window, was applied to the data along the depth axis, in order to smooth any residual spike.

Finally, the minimum/maximum global salinity value in each profile is associated to the AW/LIW core in the respective depth layer. Then, the correspondent depth and temperature values are considered.

Once the AW and LIW core are identified in each profile, the AW and LIW inter-basin variabilities were analyzed taking advantage of the boxplot approach applied to each parameter and region (Fig. 2). In Fig.2, the whiskers (black dashed line out of the box) extend to the most extreme data points not considering the outliers at the 5% significance level (pvalue≤0.05). In order to test the significance, the Student's t distribution was applied to each hydrological parameter in every sub-basin (Kreyszig and Erwin 1970). The null hypothesis (that states that the population is normally distributed) is rejected with a 5% level of statistical significance. This method is also applied to the timeseries trends. In section 3.1 we often refer to the range and skewness of the distributions, that are the difference between the upper and lower limits and the measure of the symmetry of the distributions, respectively (including only the 5% significance values).

- Considering only salinity, temperature, and depth AW and LIW values at 95% level of significance (Fig.2), as done for the spatial analysis, the timeseries from 2001 to 2019 have been computed in each subbasin to analyze the low frequency variability (LFV) and trends at interannual to decadal timescale over the available observed periods. In this respect, the high frequency variability was filtered out, first by subtracting the mean seasonal cycle to the raw timeseries, and then applying a median yearly average filter. This last step is needed since the data are not homogeneous in time in every subbasin from 2001 to 2019, and therefore without it, the seasonal variability can contaminate the estimation of the trends. The latter have been computed using the linear least-squares method to fit a linear regression model to the data."
- Therefore, with this new approach no prominence threshold has been included and only the values of salinity, temperature, and depth at 95% level of significance are considered. Deleting the outliers, any noise which can still perturb the results is ignored.
- No huge changes are found in the results except for trends over the Adriatic and Levantine subbasins, demonstrating that over them too many profiles have been ignored in the previous analyses.

**L213-218 I think this part could go into the caption of fig.2**

As the reviewer suggests, this part has been moved to the caption of Fig.2.

**L225-232 you don't give any reason in the introduction and in the list of the paper's aims on why you should look at the wavelets...**

We agree with the reviewer that the part regarding the wavelet is not complete, and more analyses would be necessary to explain the processes that are behind the variability observed in the timeseries. Therefore, a more in-depth analysis on these processes is left to future studies and the part regarding the wavelets has been removed from this manuscript.

**L236 should be "mentioned before"**

This correction has been included in the manuscript.

**L292 "more profiles"...more than what or than where?**

This part has been removed since the method for detecting the AW and LIW core has been modified.

**L301 "trend".....trend for me is something temporal, while here you are talking about spatial variations.**

This correction has been included in the manuscript.

**L340 "In the EMED....in the Tyrrhenian", but the Tyrrhenian is part of the WMED not of the EMED**

Thanks for this comment, this correction has been included in the manuscript. Please see L. 305-308.

**L354 higher than where?**

This correction has been included in the manuscript.

**L359 should be "also found by Zu et al., 2014"**

This correction has been included in the manuscript.

**L371 should be "show heterogeneous trends"**

This correction has been included in the manuscript.

**L375 unit is missing to these numbers**

This correction has been included in the manuscript.

L394 you report that LIW in the Cretan and Levantine becomes colder, which is not at all in agreement with Kassis & Korres. This discrepancy should be commented on. Furthermore a few sentences below (L397) you say something about the "warming trend in the Cretan and Levantine", so it is not clear to the reader if the LIW warms up or cools down....In addition "visual inspection" to detect such an overlapping does not seem to be an adequate statistical method to make any claim...

We thank the reviewer for the comments, and updated the manuscript with the new results and motivated the before mentioned "visual inspection". Please see L. 373-374.

L433 In the introduction of the section 3.2.2., you state that you want to discover the causes behind these trends, but then you apply a spectral analysis and in the AW and LIW section you start speaking about oscillations. This is in contrast to the rest of the paper where you have only be discussing about trends and not oscillations. Further you don't give any insights on the causes either of oscillations, nor of the trends. For these reasons I would completely eliminate the part on the wavelets, and instead try to explain the trends, even if only based on literature. Overall, neither for AW nor for LIW, this section does not give any hint to understand the causes behind the trends you described.

We agree with the reviewer that more analyses would be necessary to explain the processes that are behind the variability observed in the timeseries, and that the part regarding the wavelet is not complete. Therefore, a more in-depth analysis on these processes is left to future studies and the part regarding the wavelets has been removed from this manuscript. Therefore, we motivate that this manuscript aims only to characterize the AW and LIW on long-term timeseries, which probably constitutes the main strength of this work. Please see L. 17-20; 93-95.

**L497 when talking about these agreements, you should mention all the studies that you mentioned in the introduction as studies who address the temporal TS trends**

Thanks for the correction, we have updated the manuscript at L.419.

**L503 I guess you meant "Nile damming" not "dumping"**

Thanks for the correction, we have updated the manuscript at L.103.

**L516 abrupt shifts in which parameter? Depth?**

Yes, it is. This correction has been included in the manuscript.

**L520 the whole paper was about trends, not oscillations, and no hint on the causes of the trends is given through this analysis, which in my opinion should be removed (and maybe write a paper just on that in the future)**

The wavelet analysis has been removed and the manuscript has been updated.

**Table1: add the standard deviations in each column, not just in the last one**

This correction has been included in the manuscript.

**Fig.2 caption: replace "effective" with "retained"**

This correction has been included in the manuscript.

**REVIEWER#2**

Title could better reflect the content of the paper: Please, consider revising it to highlight the use of 20 years of data from the last decades since the use of long-term timeseries constitutes, from my viewpoint, the main strength of this work to determine trends, which have often been based of very few data)

- Suggestion 1: Characterization of the Atlantic Water and Levantine Intermediate Water in the Mediterranean Sea using last two decades of Argo Float Data
- Suggestion 2: Characterization of the Atlantic Water and Levantine Intermediate Water in the Mediterranean Sea using last 20 years of Argo Float Data

We thank the reviewer for this suggestion and update the title with the following:

"Characterization of the Atlantic Water and Levantine Intermediate Water in the Mediterranean Sea using 20 years of Argo Data".

Answers to points 2-3-4-5 are provided below:

- 2. Authors are requested to explain how this approach, based on min/max salinity peaks, is considering the positive trend due to climate change for tracking the AW and LIW. The use of geometry-based method is recommended to better detect the water masses instead of the traditional criterion based on predefined temperature and salinity ranges, as concluded by Juza et al., 2019; Vargas-Yáñez et al., 2020). In addition, they mentioned that the use of predetermined temperature and salinity ranges does not allow to detect and track spatio-temporal changes in water mass properties and may lead to erroneous characterizations and interpretations.
- 3. Authors are asked to justify the use of the minimum difference value of 0.01 for the definition of the peak.
- 4. Authors are invited to provide an explanation about the effect of the lower percentages of effective profiles related to the total number in Levantine (for AW and LIW) and in the Adriatic (for LIW) in the final results.

**5. Authors are requested to further explain how the exclusion of profiles where no peaks were found (due to intense vertical mixing) could interfere in the interpretation about the stratification trend.**

The method previously shown aimed to capture the AW and LIW cores, reducing the "noise" provided by external factors. The method considered a unique threshold for each profile and basin, provided by the prominence value 0.01. We agree with the reviewer that using different prominence values for each basin would impact less on the number of profiles considered and as consequence on the results, but the definition of the best prominence for each region would need a more in-depth study, which is not the purpose of this paper. Therefore, we choose that arbitrary value, which excluded the profiles with a less clear minimum/maximum salinity value for the AW and LIW respectively, rejecting a smaller number of profiles than other prominence values. Nevertheless, we agree that this method is too arbitrary and less robust. Therefore, we decided to modify the method, following a new strategy, discussed below. The manuscript has been updated as follows. Please see lines L183 – L213.

"A preliminary step in this analysis was the post-processing: we first applied a time sub-sampling on each profiler to obtain a more homogeneous dataset (Notarstefano and Poulain, 2009). This is applied to each float as follows: if the cycling period is 1 day or less, the profiles are sub-sampled every 5 days; if the period is 2 or 3 days, they are sub-sampled every 6 days; and if the period is 5 or 10 days, no subsampling is applied. Afterward, each profile was linearly interpolated from the surface (0 m) to the bottom every 10 m to obtain comparable profiles; and finally, a running filter with a 20 m window, was applied to the data along the depth axis, to smooth any residual spike.

Finally, the minimum/maximum global salinity value in each profile is associated to the AW/LIW core in the respective depth layer. Then, the correspondent depth and temperature values are considered. Once the AW and LIW core are identified in each profile, the AW and LIW inter-basin variabilities were analyzed taking advantage of the boxplot approach applied to each parameter and region (Fig. 2). In Fig.2, the whiskers (black dashed line out of the box) extend to the most extreme data points not considering the outliers at the 5% significance level (pvalue≤0.05). In order to test the significance, the Student's t distribution was applied to each hydrological parameter in every sub-basin (Kreyszig and Erwin 1970). The null hypothesis (that states that the population is normally distributed) is rejected with a 5% level of statistical significance. This method is also applied to the timeseries trends. In section 3.1 we often refer to the range and skewness of the distributions, that are the difference between the upper and lower limits and the measure of the symmetry of the distributions, respectively (including only the 5% significance values).

Considering only salinity, temperature, and depth AW and LIW values at 95% level of significance (Fig.2), as done for the spatial analysis, the timeseries from 2001 to 2019 have been computed in each subbasin to analyze the low frequency variability (LFV) and trends at interannual to decadal timescale over the available observed periods. In this respect, the high frequency variability was filtered out, first by subtracting the mean seasonal cycle to the raw timeseries, and then applying a median yearly average filter. This last step is needed since the data are not homogeneous in time in every subbasin from 2001 to 2019, and therefore without it, the seasonal variability can contaminate the estimation of the trends. The latter have been computed using the linear least-squares method to fit a linear regression model to the data."

- Therefore, with this new approach no prominence threshold has been included and only the values of salinity, temperature, and depth at 95% level of significance are considered. Deleting the outliers, any noise which can still perturb the results is ignored.
- No huge changes are found in the results except for trends over the Adriatic and Levantine subbasins, demonstrating that over them too many profiles have been ignored in the previous analyses.

**6. Did you exclude lower salinity in the Cretan sub-basin (i.e. where the highest salinity of LIW was found)?**

The analyses have been updated and only values considered outliers are ignored.

7. Authors are encouraged to double-check their conclusions, which seems to be in contradiction with the main results found: e.g. as it states in the results and Fig 2, LIW becomes saltier (particularly in the Adriatic) and warmer (except in the Cretan and Levantine subbasins), but the conclusion (L-487) states that LIW is less salty (i.e. fresher), and colder.

We thank the reviewer for this comment, but those conclusions refer to the mean zonal gradients over (inter-basin variability) the Mediterranean Sea at Lines 420-423, which is also captured by Fig.2.

**8. It may also be worth further investigation about the drivers of the AW and LIW interbasin and interannual variability in the context of this study.**

We agree with the reviewer that more analyses would be necessary to explain the processes that are behind the variability observed in the timeseries, and that the part regarding the wavelet is not complete. Therefore, a more in-depth analysis on these processes is left to future studies and the part regarding the wavelets has been removed from this manuscript.

- 9. As a general comment regarding the in-text citation and reference list order:
- please consider to review the in-text citation order, listing the sources alphabetically by author surname (APA reference) or using author-date referencing, ordering the citation based on data of publication.

We thank the reviewer for this suggestion, we have sorted the publications in the right alphabetical order.

please, consider to review the reference list, to arrange entries in the alphabetical order by the surname of the fist author (e.g. incorrect order found in L555 before L558; L571 after L565; L625 before L629, etc.).

Thanks for this comment, we have sorted the publications in the right alphabetical order.

**Technical corrections**

L-56: specify the different approaches for identifying AW and LIW and the advantage of the methodology used in this study by means of the identification of min and max values, particularly for the characterization of the AW (i.e. much more influenced by other forcings)

We thank the reviewer for the comment. This part has been removed in this version of the manuscript.

**L-56: add references of different approaches (e.g. Juza et al., 2019; Várgas-Yañez et al., 2020)**

Thanks, these references have been included.

**L-65: add references about the hydrological properties of AW (as the authors properly do for the LIW in L-77)**

Thanks, these references have been included.

**L-84: add additional recent studies including, for example, Juza et al., 2019; Várgas-Yañez et al., 2020**

Thanks, these references have been included.

L-143: add one paragraph about the new initiatives of deploying floats in shallow coastal areas (e.g. off the North Adriatic coast and off the Bulgarian Black Sea ). Authors can access to the tracks in this link: https://www.euro-argo.eu/EU-Projects/Euro-Argo-RISE-2019-2022/Access-to-Euro-Argo-RISE-Data

We thank the reviewer for this suggestion, but we believe it is out of topic and therefore we don't include this in the manuscript.

**L-151: please, add the URL to access the float monitoring and data (https://fleetmonitoring.euro-argo.eu/dashboard?Status=Active)**

Thanks per the suggestion, this link has been included. Please see L.150.

**L-235: please, add eight Mediterranean "climatic" regions**

This correction has been included in the manuscript.

**L-345: authors are encouraged to justify the main reasons of the contradiction found with the results provided by Millot (2007), is it just a matter of the period selected?**

This correction has been included in the manuscript. Please see L.314.

**L-345: do you mean Mediterranean "outflow"?**

Yes, we did. This correction has been included in the manuscript. Please see L.313.

**L-365: authors are encouraged to further explain the agreement found with Schuckmann et al., (2019) about the SST warming (0.018 versus 0.04 °C/yr., it is more than twice)**

Thanks for the comment, we give explanation at L. 337-341.

**L-469: please, consider to replace non-stationary with "variable".**

This correction has been included in the manuscript.

**Fig.1: please, increase resolution of the image. The use of different colors for different subbasins might help.**

Thanks for the suggestion, we increased the resolution and changed the colors.

**Fig.3-8: authors are invited to include the trends/yr. included in Table 1 in each one of the subplots, to better identify higher/lower trends in the figures.**

We thank the reviewer for the suggestion but we believe that the figure is cleaner as it is, and Table 1 is complementary with the "trend plots".

**Fig.5 and 8: In order to help in the interpretation of the results at a first glance, please include the following in the caption of the figures. "Positive/negative trends (red/blue squares) in this case corresponds to an increase/decrease of the depth (i.e. deeper/shallower)"**

Thanks for the suggestions. These comments have been included in their captions.

---

## Author Response (AR2)

**AUTHOR'S RESPONSE**

**REVIEWER#1**

We thank the reviewer for the comments and suggestions provided in the review process, which contributed to improve the quality of this paper.

**REVIEWER#2**

We thank the reviewer for the comments and suggestions. A detailed answer to the reviewer is provided below. In bold the reviewer's comments are shown. The updates in the manuscripts are in red font.

**- Lines 280-284. LIW bimodality in the depth domain is something also described and explained by Mihanović et al. (2021), it should be also commented here.**

Thanks for the comment, we have updated the manuscript at L.283-290.

**- Lines 359-360. The Adriatic salinity trends are about 5 times larger than the Adriatic salinity trends observed in the period 1952-2010 and quantified by Vilibić et al. (2013). This should be commented, also the possible source of such a large trends, which largely come by the biasing of the signal by the BiOS oscillations (see comment below).**

Thanks for the comment, we have updated the manuscript at L.367-373.

**- In all places where Kokkini et al. (2019) is used for commenting the Southern Adriatic thermohaline properties, please use also Mihanović et al. (2021), at it presents these properties in the quoted region but over a longer timescale.**

Thanks for the comment, we have updated the manuscript as suggested.

**- It would be appropriate in the bulleted conclusions part to added a bullet discussing in general the impact of time series length to the trend estimates. E.g. in the Adriatic, the salinity trends are clearly impacted by the BiOS, as the series starts in 2008, when the anticyclonic BiOS was present, so that the salinity trends are so large. Here, the periodicity of the BiOS is of the same order of magnitude than the length of the time series in the Adriatic. That might be also relevant for other sub-regions.**

Thanks for the comment. A new bulleted conclusion, which emphasizes "**the impact of time series length to the trend estimates**", has been included at L.466-470.

**- Please add Vilibić et al. (2012 and 2020) to the reference list.**

Thanks for the comment, Vilibić et al. (2012 and 2020) have been added to the reference list.